# Metal-Cation-Induced Tiny Ripple on Graphene

**DOI:** 10.3390/nano14191593

**Published:** 2024-10-02

**Authors:** Yingying Huang, Hanlin Li, Liuyuan Zhu, Yongshun Song, Haiping Fang

**Affiliations:** 1School of Physics, East China University of Science and Technology, Shanghai 200237, China; li_lihanlin0627@126.com (H.L.); zhuly666@126.com (L.Z.); fanghaiping@sinap.ac.cn (H.F.); 2School of Physics, Zhejiang University, Hangzhou 310027, China

**Keywords:** tiny ripple, graphene, metal cation, electronic property

## Abstract

Ripples on graphene play a crucial role in manipulating its physical and chemical properties. However, producing ripples, especially at the nanoscale, remains challenging with current experimental methods. In this study, we report that tiny ripples in graphene can be generated by the adsorption of a single metal cation (Na^+^, K^+^, Mg^2+^, Ca^2+^, Cu^2+^, Fe^3+^) onto a graphene sheet, based on the density functional theory calculations. We attribute this to the cation–π interaction between the metal cation and the aromatic rings on the graphene surface, which makes the carbon atoms closer to metal ions, causing deformation of the graphene sheet, especially in the out-of-plane direction, thereby creating ripples. The equivalent pressures applied to graphene sheets in out-of-plane direction, generated by metal cation–π interactions, reach magnitudes on the order of gigapascals (GPa). More importantly, the electronic and mechanical properties of graphene sheets are modified by the adsorption of various metal cations, resulting in opened bandgaps and enhanced rigidity characterized by a higher elastic modulus. These findings show great potential for applications for producing ripples at the nanoscale in graphene through the regulation of metal cation adsorption.

## 1. Introduction

Graphene, as a two-dimensional material, possesses superior characteristics, including the highest room-temperature carrier mobility and high mechanical strength [1,2]. Due to its flexibility, graphene exhibits no resistance to out-of-plane deformations [3,4]. Lattice deformations induced by surface corrugation, such as ripples, can significantly modify the physical properties and chemical reactivity of graphene [5,6], including hydrogen splitting [7], strong Raman enhancement [8], pseudomagnetic fields [9,10], bandgap opening [11,12,13,14,15,16], carrier scattering [17,18], and electron–hole puddles [19,20,21]. Therefore, controlling graphene ripples becomes a powerful method for tuning the physical properties and chemical reactivity of this ultimate thin film [7,8,9,12,16,22,23,24,25].

Corrugation in graphene can be formed using various methods, such as spontaneous dynamic ripples of single graphene [26,27], defect-induced and doping methods [28,29], the growing of graphene on metal substrates [30], and graphene transfer processes [31,32]. The formation of ripples at the nanoscale or smaller can create a very narrow bandgap in graphene, which is crucial for practical applications in graphene-based nanoelectronics and nanoelectromechanical devices. However, it is difficult to produce ripples smaller than one nanometer using these current methods, and accurately manipulating these ripples to achieve the desired properties remains a big challenge [5,33].

The cation–π interaction, a type of non-covalent interaction formed between cations and π-electron-rich carbon-based structures [34], provides a versatile and reversible approach to modifying physical properties without altering the underlying chemical structures [35]. The adsorption of cations on graphene has been observed in both gas and solvent phases [36,37,38,39]. However, no studies have investigated the effect of cation adsorption on the ripple formation in graphene. By controlling the cation adsorption site, the formation of ripples can be manipulated. Additionally, the type of cation is expected to affect the size of the ripples formed due to variations in the strength of the cation–π interactions.

In this study, by simulating the adsorption behavior of various metal cations, including Na^+^, K^+^, Mg^2+^, Ca^2+^, Cu^2+^, and Fe^3+^, on the graphene sheets, we find that tiny ripples in graphene can be induced by the adsorption of metal cations. This ripple formation is attributed to the cation–π interaction between the metal cation and the aromatic rings on the graphene surface, which makes the carbon atoms closer to metal ions, causing deformation of the graphene sheet, especially in the out-of-plane direction, namely ripples. The equivalent pressures applied to graphene sheets, which are induced by cation adsorption, are on the magnitude of GPa. These ripples modify the electronic and mechanical properties of graphene, leading to a non-zero bandgap and enhanced rigidity characterized by a higher Young’s modulus. These findings offer a new perspective for understanding and controlling the properties of graphene by inducing tiny ripples through metal cation adsorption.

## 2. Methods

The geometry optimizations of various cations (Na^+^, K^+^, Mg^2+^, Ca^2+^, Cu^2+^, Fe^3+^) adsorbed on a graphene sheet were performed using density functional theory (DFT) [40,41] calculations, implemented in the Vienna ab initio Simulation Package (VASP 5.4.4) [42]. The lattice parameters of the graphene sheet were a = 8.55 Å and b = 7.40 Å. The electron–ion interactions and exchange–correlation functions were described using the projector augmented wave (PAW) method [43] and the generalized gradient approximation (GGA) with the Perdew–Burke–Ernzerhof (PBE) functional [44], respectively. The electron wavefunction was expanded in a plane-wave basis with a cutoff energy of 1000 eV. Van der Waals corrections were accounted for using the DFT-D3 model proposed by Grimme [45]. The convergence criteria for energy and force were set to 10^−4^ eV and 10^−2^ eV Å^−1^, respectively. The Brillouin zones were sampled using k-point grids with a uniform spacing of 0.25 Å^−1^. For Cu^2+^ and Fe^3+^, which have unpaired electrons, spin polarization calculations were performed. For band structure calculations, the Brillouin zone was sampled using VASPKIT [46]. To reliably calculate the band structure, the hybrid functional HSE06 was applied [47,48]. A vacuum region of over 20 Å was set to eliminate the interactions between layers.

To analyze the lowest unoccupied molecular orbital (LUMO) energies of Mg^2+^, Ca^2+^ and Cu^2+^, the DFT calculations were performed using the Gaussian 09 software package [49]. All geometry optimizations and frequency calculations were conducted at the B3LYP/SDD level of theory [50]. The LUMO equivalent surface was plotted with the Multiwfn program 3.8 (dev) [51].

## 3. Results and Discussion

To study the adsorption behavior of various metal cations, including Na^+^, K^+^, Mg^2+^, Ca^2+^, Cu^2+^, and Fe^3+^, on a graphene sheet, we initially constructed structural models by placing different cations at various sites on a graphene sheet consisting of twenty-four carbon atoms. After full optimization based on DFT calculations, the geometric structures of the various metal cations absorbed on graphene are illustrated in Figure 1a. All the metal cations occupy the hollow sites of the graphene sheet. The z-direction distances (*d*) between Na^+^, K^+^, Mg^2+^, Ca^2+^, Cu^2+^, and Fe^3+^ and the graphene sheet are 2.51 Å, 2.97 Å, 2.13 Å, 2.39 Å, 1.98 Å, and 1.68 Å, respectively (Figure 1b). When comparing cations within the same valence state, Na^+^ is much closer to the graphene surface than K^+^, and Mg^2+^ is closer to the graphene surface than Ca^2+^. This observation aligns with the smaller atomic radii of Na^+^ and Mg^2+^ compared to K^+^ and Ca^2+^, respectively. Additionally, when comparing metal ions within the same period, Mg^2+^ exhibits a shorter adsorption distance than Na^+^, while Ca^2+^ is closer than K^+^. This trend can be attributed to the higher charges of Mg^2^⁺ and Ca^2^⁺ relative to Na⁺ and K⁺, respectively. Although the atomic radius of Fe^3+^ is comparable to that of Mg^2+^, its higher charge results in a shorter adsorption distance. Furthermore, Cu^2+^ demonstrates a shorter adsorption distance than both Ca^2+^ and Mg^2+^, likely due to the oxidative capability of copper ions.

When a metal cation is adsorbed on the graphene surface, the ion adsorption energy (*E*_i_) can be calculated using the following formula:(1)$$E_{i}=E_{ion@G}-E_{ion}-E_{G}$$
where *E*_ion_, *E*_G_, and *E*_ion@G_ are the total energies of the isolated ion, the graphene sheet, and the ion-adsorbed graphene sheet, respectively. As shown in Figure 1c, the ion adsorption energies (*E*_i_) for the systems with adsorption of Na^+^, K^+^, Mg^2+^, Ca^2+^, Cu^2+^, and Fe^3+^ are −51.4, −47.7, −170.9, −142.5, −307.3, and −675.6 kcal/mol, respectively. The magnitude of *E*_i_ is consistent with previous results [37]. The order of adsorption energy for these cations exhibits an inverse relationship with the varying trend in *d*, indicating that a smaller distance between the metal cation and the graphene sheet corresponds to a stronger adsorption effect.

The oxidative property of Cu^2+^ arises energies (Cu^2+^ > Mg^2+^ > Ca^2+^).

Furthermore, the electron transfer between the metal cation and the graphene sheet was explored. The numbers of electrons transferred from the graphene sheet to the unoccupied valence orbits of Na^+^, K^+^, Mg^2+^, Ca^2+^, Cu^2+^, and Fe^3+^ are 0.053, 0.050, 0.389, 0.359, 1.119, and 1.677 e, respectively (Figure 1d). We found that the adsorption energies of various metal cations on the graphene sheets are directly proportional to the number of electrons transferred from the graphene sheet to the corresponding metal cations. Thus, the metal cations with higher valence states tend to accept more electrons, indicating a stronger coupling interaction between their unoccupied valence orbitals and the graphene.

As shown in Figure 2, in the partial electron density of states (DOS) near the Fermi level of Cu^2+^@graphene and Fe^3+^@graphene, the 3D-orbital densities are considerably higher than those of other orbitals, and overlap with the DOS of carbon atoms at some peaks, indicating that the interactions between electrons in the 3d orbitals and carbon atoms help the effective adsorption of transition metal cations on graphene sheets.

The adsorption of a metal cation on a graphene sheet can induce a tiny ripple. To characterize this ripple, we define two parameters: the deformation in the z direction (Δ*Z*) and the relative area change in the rippled graphene sheet compared to flat graphene (Δ*S*). These parameters help describe the geometric scale of the ripple produced by metal cation adsorption. The values of Δ*Z* corresponding to rippled graphene sheets induced by the adsorption of Na^+^, K^+^, Mg^2+^, Ca^2+^, Cu^2+^, and Fe^3+^ are 0.019 Å, 0.006 Å, 0.096 Å, 0.080 Å, 0.079 Å, and 0.053 Å, respectively (Figure 3a). Importantly, the values of Δ*S* for rippled graphene sheets induced by various metal cations are all less than 1%. Our findings indicate that divalent and tervalent cations tend to produce larger ripples, which can be attributed to the stronger cation–π interactions between the cations and the graphene sheets.

When a cation is adsorbed onto a graphene sheet, the previously flat graphene structure becomes destabilized, as it can no longer maintain its lowest free energy state. The delocalization of cation–π interactions brings the carbon atoms on the graphene closer to the metal ion, causing the graphene sheet to bend. Thus, the deformation, particularly in the out-of-plane direction, namely the ripple, occurs as a direct result of the adsorption.

To quantitatively express the adsorption effect of cations on graphene, we introduce the concept of equivalent pressure *P*. Using Taylor’s expansion, we can express the energy associated with small z-direction deformations up to the second order as follows:(2)$$E\left(z\right)=E\left(0\right)+\frac{1}{2}\beta z^{2}$$

Here, $E\left(0\right)\;$ and $E\left(z\right)$ represent the total energies of the graphene sheet before and after cation adsorption, respectively. The equivalent force *F* exerted by the cation on the graphene can then be expressed as
(3)$$F=\frac{dE}{dz}\approx \beta z$$

Consequently, the equivalent pressure *P* is defined as
(4)P=FS=βzS=2∆EzS
where $∆E=E\left(z\right)-E(0)$. The area *S* is considered constant, given the minimal deformation in the x and y directions (Δ*S* < 0.75%; see Appendix A). The values of ∆E are presented in Appendix A, and the equivalent pressures for different metal cations are shown in Figure 3b. The equivalent pressures for Na^+^@graphene, K^+^@graphene, Mg^2+^@graphene, Ca^2+^@ graphene, Cu^2+^@graphene, and Fe^3+^@graphene are 2.24 GPa, 2.44 GPa, 4.79 GPa, 3.03 GPa, 5.11 GPa, and 8.70 GPa, respectively. Notably, the varying trend in equivalent pressure closely follows the order of adsorption energies of the various metal cations on graphene sheets. Previous studies have applied continuum theory to investigate corrugation in graphene [33,52,53]. When the corrugation is isotropic (i.e., forming ripples), the total energy of the system aligns with Taylor’s expansion, indirectly validating the approach and formulas used in this study to calculate equivalent pressure.

The mechanical properties of rippled graphene sheets with the adsorption of various metal cations show notable changes in their Young’s modulus. The calculated values for Na^+^@graphene, K^+^@graphene, Mg^2+^@graphene, Ca^2+^@ graphene, Cu^2+^@graphene, and Fe^3+^@graphene are 325.53 N/m, 325.83 N/m, 342.15 N/m, 330.84 N/m, 340.86 N/m, and 338.81 N/m, respectively (Table 1). Compared to graphene without metal cation adsorption, the elastic modulus of the rippled graphene increases with the presence of these cations. This enhancement in rigidity suggests that the slight rippling induced by metal cation adsorption improves the overall mechanical properties of graphene.

In flat graphene without metal cation adsorption, the unique electronic structure gives rise to a Dirac cone within the Brillouin zone. At the Γ point, the conduction band and the valence band converge, resulting in a zero bandgap (0 eV). In contrast, based on hybrid functional HSE06, the calculated band gaps for rippled graphene sheets induced by Na^+^, K^+^, Mg^2+^, Ca^2+^, Cu^2+^, and Fe^3+^ are 0.031 eV, 0.041 eV, 0.020 eV, 0.032 eV, 0.118 eV, and 0.136 eV, respectively (Figure 4b and Appendix A). The adsorption of these metal cations leads to the opening of energy bandgaps in graphene sheets, indicating a transition from conductors with zero bandgap to semiconductors with non-zero bandgaps. This demonstrates that the slight rippling of graphene, induced by weak cation–π interactions, can effectively regulate its energy band structure. Consequently, the change in bandgap can be used to characterize the structural damage of graphene after deformation, as a previous study reported an increase in bandgap when carbon atoms are extracted from a graphene sheet in the z-direction [15].

## 4. Conclusions

In summary, the tiny out-of-plane ripples on graphene sheet induced by various metal cations have been achieved by adsorbing various metal cations such as Na^+^, K^+^, Mg^2+^, Ca^2+^, Cu^2+^, and Fe^3+^ based on DFT simulations. The formation of the ripple is primarily attributed to the cation–π interactions between the metal cations and graphene, which brings the carbon atoms closer to metal ions, inducing deformation of the graphene sheet, referred to as the ripple. The equivalent pressures exerted on graphene sheets in the out-of-plane direction, generated by these cation–π interactions, reach the magnitude of GPa. More importantly, the adsorption of metal cations not only enhances the rigidity of rippled graphene sheets but also opens their bandgaps.

Notably, the tiny ripple on graphene is induced only by a single metal cation. With stronger adsorption from multiple cations, we anticipate that the ripple or deformation of graphene will significantly increase. Therefore, we aim to explore the possibility of obtaining a graphene sheet with a wide bandgap by adsorbing multiple metal cations—this will be a primary focus for our future research. These findings highlight the significant potential to regulate nanoscale ripples in graphene by controlling the cation adsorption, leveraging the weak non-covalent cation–π interactions.

## Figures and Tables

**Figure 1 nanomaterials-14-01593-f001:**
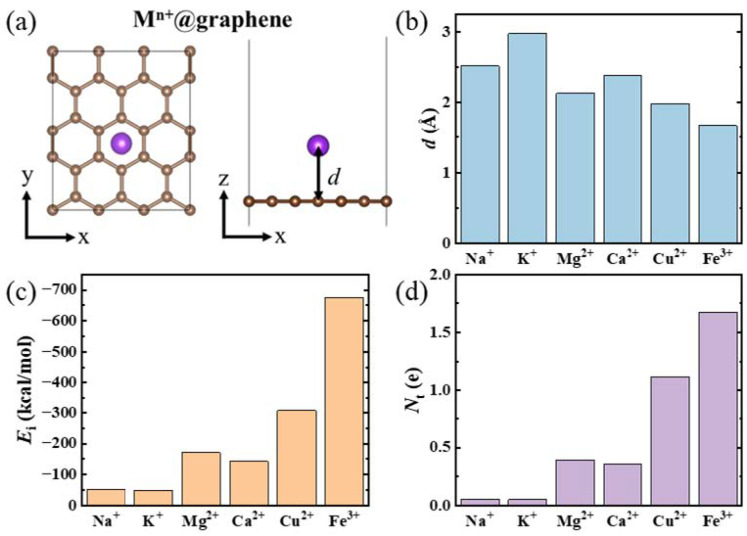
(**a**) Structural configuration of a metal cation adsorbed on a graphene sheet. M^n+^ denotes the metal cation, including Na^+^, K^+^, Mg^2+^, Ca^2+^, Cu^2+^, and Fe^3+^. (**b**) Distances between various metal cations and the corresponding graphene sheets. (**c**) Adsorption energies of the various metal cations on the graphene sheet. (**d**) Number of electrons transferred from the graphene sheet to the metal cations.

**Figure 2 nanomaterials-14-01593-f002:**
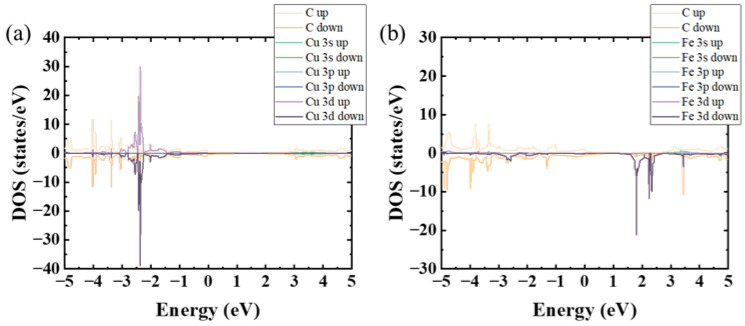
Partial electron density of states near the Fermi level of (**a**) Cu^2+^@graphene and (**b**) Fe^3+^@graphene.

**Figure 3 nanomaterials-14-01593-f003:**
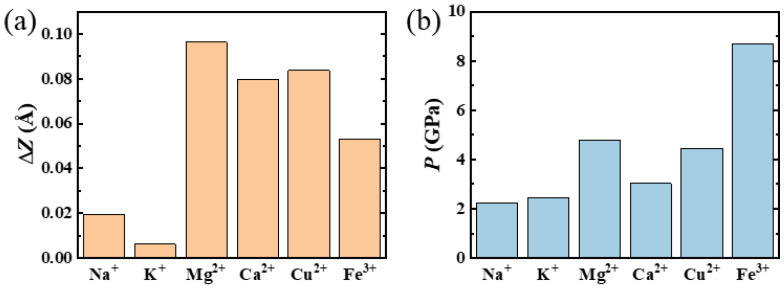
(**a**) Out-of-plane deformation (Δ*Z*) of rippled graphene induced by the adsorption of various metal cations. (**b**) Equivalent pressure (*P*) exerted by the metal cation on the graphene sheet.

**Figure 4 nanomaterials-14-01593-f004:**
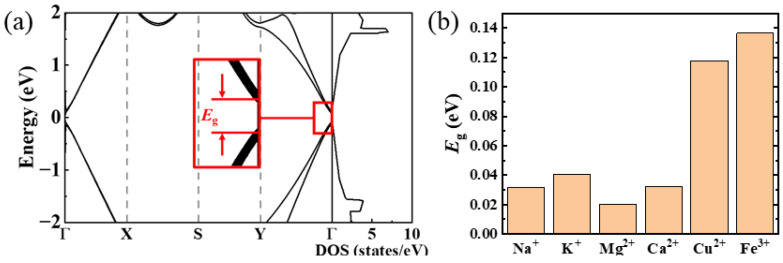
(**a**) Band structure and density of states (DOS) for Fe^3+^ adsorbed on the graphene sheet. *E*_g_ represents the band gap. (**b**) Band gaps of rippled graphene sheets induced by various metal cations.

**Table 1 nanomaterials-14-01593-t001:** Elastic tensors and Young’s modulus of 2D rippled graphene sheets with adsorption of various cations.

System	C11 (N/m)	C12 (N/m)	C22 (N/m)	C33 (N/m)	Young’s Modulus (N/m)
Graphene	325.74	78.16	347.47	138.87	325.39
Na+@graphene	326.09	76.88	347.11	138.92	325.53
K+@graphene	326.41	77.29	347.61	138.98	325.83
Mg2+@graphene	342.73	64.06	357.12	148.78	342.15
Ca2+@graphene	333.94	72.99	350.89	140.35	330.84
Cu2+@graphene	344.97	60.50	362.21	143.20	340.86
Fe3+@graphene	350.08	68.56	352.71	142.53	338.81

## Data Availability

The datasets generated and/or analyzed during the current study are available from the corresponding author on reasonable request.

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
