# Peer review of "Metal-Cation-Induced Tiny Ripple on Graphene"

_nanomaterials, 2024, doi:10.3390/nano14191593_

Round 1

Reviewer 1 Report

Comments and Suggestions for Authors

The authors have taken an approach by introducing undulations in the graphene structure to study its electronic properties. However, the method of adsorbing cationic metals onto a flat graphene surface is something that has been well-explored in previous research. Similar effects have also been observed when macromolecules are adsorbed. Because of this, the study doesn’t really bring new or original insights to the table. Additionally, the bandgap opening of up to 50 meV observed with Cu2+ is quite small and unlikely to be useful for practical electronic applications. Similar small bandgaps have been seen before in studies on quantum Hall effects.

The manuscript could benefit from a clearer explanation of the research problem and how the authors intend to solve it. While the DFT calculations are standard, it might be more effective to analyze the potential effects of local symmetry breaking where the element is adsorbed using the GW method, especially in terms of bandgap. The results and discussions don’t fully explain why the observed effects are happening; they mainly focus on binding energy without diving deeper into other possible reasons.

It would be helpful if the authors could present the data in its original forms, like showing the band structure, density of states, and not only histograms. Additionally, it’s surprising that the authors didn’t try adding more elements to the graphene or increasing their concentration to create more pronounced ripples, which might have led to bandgap values more relevant to current electronic technology, as has been done with graphene nanoribbons.

Comments on the Quality of English Language

English can be improved. Typos here and there 

Author Response

Referee 1:

Comments and Suggestions for Authors:

The authors have taken an approach by introducing undulations in the graphene structure to study its electronic properties.

Reply: We sincerely thank the referee for these constructive suggestions.

(1) However, the method of adsorbing cationic metals onto a flat graphene surface is something that has been well-explored in previous research. Similar effects have also been observed when macromolecules are adsorbed. Because of this, the study doesn’t really bring new or original insights to the table.

Reply: We thank the referee for his/her comments.

We have investigated the research on the adsorption of cationic metals onto a graphene or graphene oxide surface. The previous works are mainly concerned with the research about the effects of cation adsorption onto graphene on the colloidal properties and joint toxicity [1], the mechanism of adsorption and coadsorption of organic pollutants and a heavy metal by graphene oxide and reduced graphene materials [2], ion enrichment on the hydrophobic carbon-based surface in aqueous salt solutions [3], and the effects of adsorption of aromatic macromolecules onto graphene on the entropy-tailored behavior [4].

In this work, we focused on the research of structural properties of graphene, particularly the ripples, induced by the adsorption of single metal cation. This is important, because the control of the size of the ripples on graphene can significantly modify the physical properties and chemical reactivity of the graphene [5-6]. Our findings indicate that the tiny ripple on graphene surface can be induced by a single metal cation, which is attributed to the cation-π interaction between the metal cation and the aromatic rings on the graphene surface. This makes the carbon atoms closer to metal ions, causing the generation of the deformation of graphene sheet especially in out-of-plane direction, namely the ripple. More importantly, the electronic and mechanical properties of rippled graphene sheets are modified by the adsorption of metal cation, resulting in an opened bandgap and enhanced rigidity characterized by a higher elastic modulus. These findings show great potential for the applications of producing ripples at the nanoscale in graphene by the regulation of metal cation adsorption.

(2) Additionally, the bandgap opening of up to 50 meV observed with Cu2+ is quite small and unlikely to be useful for practical electronic applications. Similar small bandgaps have been seen before in studies on quantum Hall effects.

Reply: We thank the referee for his/her comments.

Given that the PBE functionals typically underestimates the bandgap, we have verified the bandgaps of rippled graphene sheets using a more reliable hybrid functional HSE06. As shown in Table S1, the band gaps of rippled graphene sheets induced by a Na+, K+, Mg2+, Ca2+, Cu2+ and Fe3+ are 0.031 eV, 0.041 eV, 0.020 eV, 0.032 eV, 0.118 eV, and 0.136 eV, respectively. Furthermore, considering the multi-ion effects, the induced ripples will be significantly enlarged and the corresponding bandgaps can be further increased, which will be useful for practical electronic application.

Table R1. Band gaps (eV) of rippled graphene sheets induced by a metal cation.

Figure R1. (a) Band structure and density of states (DOS) for the Fe3+ adsorbed on graphene sheet. Eg represents the bandgap. (b) Bandgaps (Eg) of rippled graphene sheets induced by various metal cations.

Changes made:

We have modified the figures (Figure 3) as the same with Figure R1 and discussion of bandgaps in the revised manuscript:

“In contrast, the calculated bandgaps based on hybrid functional HSE06 for rippled graphene sheets induced by Na+, K+, Mg2+, Ca2+, Cu2+, and Fe3+ are 0.031 eV, 0.041 eV, 0.020 eV, 0.032 eV, 0.118 eV, and 0.136 eV, respectively (see Figure 3b). The adsorption behaviors of Na+, K+, Mg2+, Ca2+, Cu2+ and Fe3+ on graphene lead to opened energy bandgaps of graphene sheets, indicating a transition from conductors with zero bandgap to semiconductors with nonzero bandgaps.”

(3) The manuscript could benefit from a clearer explanation of the research problem and how the authors intend to solve it. While the DFT calculations are standard, it might be more effective to analyze the potential effects of local symmetry breaking where the element is adsorbed using the GW method, especially in terms of bandgap.

Reply: We thank the referee for his/her comments.

Since the calculation of bandgap using the GW method need a large amount of computation, within the limited time for revised manuscript we have chosen the relatively reliable hybrid functional HSE06 with acceptable amount of computation. The results are shown in Table R1 and Figure R1. Obviously, the PBE functional underestimates the bandgaps, while the HSE06 functional can better describe the bandgaps of rippled graphene sheets induced by various metal cations.

(4) The results and discussions don’t fully explain why the observed effects are happening; they mainly focus on binding energy without diving deeper into other possible reasons.

Reply: We thank the referee for his/her comments.

We have thoroughly explained the mechanism for the generation of ripple. When a cation is adsorbed on graphene sheet, the flat graphene will be disrupted to maintain the lowest free energy because the flat sheet is no longer stable with the cation adsorption. The delocalization of cation-π interaction between the cation and the graphene sheet makes the carbon atoms on the graphene closer to the metal ion, causing that the graphene sheet tends to bend. Thus, the deformation especially in out-of-plane direction, namely the ripple, is produced.

Changes made:

We have added discussion on the reason for the ripple generation in the revised manuscript:

“When a cation is adsorbed on graphene sheet, the flat graphene sheet will be disrupted to maintain the lowest free energy because the flat sheet is no longer stable with the cation adsorption. The delocalization of cation-π interaction between the cation and the graphene sheet makes the carbon atoms on the graphene closer to the metal ion, causing that the graphene sheet tends to bend. Thus, the deformation especially in out-of-plane direction, namely the ripple, is produced.”

(5) It would be helpful if the authors could present the data in its original forms, like showing the band structure, density of states, and not only histograms.

Reply: We thank the referee for his/her constructive suggestions.

We have added the pictures of density of states, which are shown in Figure R2.

Figure R2. (a)-(f) Band structures and density of states (DOS) for rippled graphene sheets induced by Na+, K+, Mg2+, Ca2+, Cu2+, and Fe3+, respectively.  Eg represents the bandgap.

Changes made:

We have added the pictures of density of states in Figure 3 as the same with Figure R1a and Figure S3 as the same with Figure R2 in the revised manuscript and revised supporting information.

(6) Additionally, it’s surprising that the authors didn’t try adding more elements to the graphene or increasing their concentration to create more pronounced ripples, which might have led to bandgap values more relevant to current electronic technology, as has been done with graphene nanoribbons.

Reply: We thank the referee for his/her comments.

In this study, we focused on the effects of a single cation adsorption in different types on the ripple in the graphene surface, which is just our primary study on the effects of ion adsorption on graphene ripples, and achieved tiny ripples. The next study projects are the effects of the adsorption of other elements including unconsidered cations in this work and all anions, and ion concentrations, and multi-ion effects on graphene ripples, which can create more pronounced ripples. Therefore, the effects of other elements and high ion concentrations and multi-ion effects on graphene ripples are beyond the scope of this work and will be reported in future studies.

 (7) Comments on the Quality of English Language

English can be improved. Typos here and there.

Reply: We thank the referee for his/her comments.

We have carefully checked for all grammatical errors and typos, and made the revision in the revised manuscript:

  1. Line 3-5 in the part of Abstract, “In this study, using the density functional theory simulations of attaching a metal cation (Na+, K+, Mg2+, Ca2+, Cu2+, Fe3+) to the graphene sheet, the tiny ripple in graphene is produced.” is changed to “In this study, we report that a tiny ripple in graphene can be produced by the adsorption of a single metal cation (Na+, K+, Mg2+, Ca2+, Cu2+, Fe3+) onto graphene sheet based on the density functional theory”
  2. Line 5-6 in the part of Abstract, “We attribute this ripple to the cation-π interaction between the metal cation and the aromatic rings on the graphene surface” is changed to “We attribute this to the cation-π interaction between the metal cation and the aromatic rings on the graphene surface, which makes the carbon atoms closer to metal ions, causing the generation of the deformation of graphene sheet especially in out-of-plane direction, namely the ripple.”
  3. Line 10-12 in the part of Abstract is changed to “More importantly, the electronic and mechanical properties of graphene sheets are modified by the adsorption of various metal cations, resulting in opened bandgaps and enhanced rigidities characterized by the higher elastic modulus.”.
  4. Line 4 of the second paragraph in the part of Introduction: “wavelengths” is changed to “sizes”.
  5. Line 7 of the second paragraph in the part of Introduction: “is” is changed to “belongs to”.
  6. Line 11-12 of the second paragraph in the part of Introduction: “the type of cation should influence the ripples formed on graphene though varying the cation-p” is changed to “the type of cation should influence the ripple size formed on graphene via varying the strength of cation-p interaction.”.
  7. Line 2-3 of the second paragraph in the part of Introduction is changed to “a tiny ripple in graphene can be induced by the adsorption of metal cation.”.
  8. Line 3-8 of the third paragraph in the part of Introduction is changed to “This ripple formation is attributed to the cation-p interaction between the metal cation and the aromatic rings on the graphene surface, which makes the carbon atoms closer to metal ions, causing the generation of the deformation of graphene sheet especially in out-of-plane direction, namely the ripple. The equivalent the pressures applied to graphene sheets, which are induced by cation adsorption, are on the magnitude of GPa.”.
  9. Line 11 of the third paragraph in the part of Introduction is changed to “controlling the properties of graphene by inducing tiny ripples with metal cation adsorption.”.
  10. Line 1-2 in the part of Method: “optimization” and “was” are changed to “optimizations” and “were”, respectively.
  11. Line 11 of the first paragraph in the part of Results and discussion: “For” is changed to “As for”.
  12. Line 13-14 of the first paragraph in the part of Results and discussion is changed to “This is accessible because Mg²⁺ and Ca²⁺ carry more charges compared to Na⁺ and K⁺, respectively.”
  13. Line 15 of the first paragraph in the part of Results and discussion: “minimal” is changed to “shorter”.
  14. Line 11 of the second paragraph in the part of Results and discussion: “with the distance d” is changed to “with the varying trend of d”.
  15. Line 5 of the fourth paragraph in the part of Results and discussion: “sheet” and “number” are changed to “sheets” and “numbers”, respectively.
  16. Line 1 of the sixth paragraph in the part of Results and discussion: “effect” is deleted.
  17. Line 5 of the sixth paragraph in the part of Results and discussion: “with” is changed to “induced by”.
  18. Line 7 of the sixth paragraph in the part of Results and discussion: “graphene sheets with” is changed to “rippled graphene sheets induced by”.
  19. Line 10 of the sixth paragraph in the part of Results and discussion: “graphene” is changed to “graphene sheets”.
  20. Line 8-10 of the ninth paragraph in the part of Results and discussion is changed to “Notably, the varying trend of equivalent pressure closely follows the order of adsorption energies of various metal cations on graphene sheets.”.
  21. Line 1 of the tenth paragraph in the part of Results and discussion is changed to “The mechanical properties of rippled graphene sheets with the adsorption of various metal cations were analyzed”.
  22. The last sentence of the last paragraph in the part of Results and discussion is changed to “Therefore, the change in bandgap can be used to characterize the structural damage of graphene after deformation, as the previous study reported that the bandgaps is increased when carbon atoms are extracted from a graphene sheet in the z-direction [15].”
  23. Line 4-7 of the first paragraph in the part of Conclusion is changed to “which makes the carbon atoms closer to metal ions, causing the generation of the deformation of graphene sheet, namely the ripple. The equivalent pressures applied to graphene sheets in out-of-plane direction, generated by metal cation-π interactions, reache the magnitude of GPa.”.
  24. Line 3 of the second paragraph in the part of Conclusion, the word “significantly” is added.

References

R1. Gao, Y.; Ren, X.; Wu, J.; Hayat, T.; Alsaedi, A.; Cheng, C.; Chen, C. Graphene oxide interactions with co-existing heavy metal cations: adsorption, colloidal properties and joint toxicity. Environ. Sci.: Nano 2018, 5, 362-371.

R2. Wang, J.; Chen, B. Adsorption and coadsorption of organic pollutants and a heavy metal by graphene oxide and reduced graphene materials. Chem. Eng. J. 2015, 281, 379-388.

R3. Shi, G.; Liu, J.; Wang, C.; Song, B.; Tu, Y.; Hu, J.; Fang, H. Ion enrichment on the hydrophobic carbon-based surface in aqueous salt solutions due to cation-p interactions. Sci. Rep. 2013, 3, 3436.

R4. Meng, C.; Gao, K.; Tang, S.; Zhou, L.; Lai, W.; Luo, L.; Wang, X.; Liu, Y.; Wang, K.; Chen, Y.; et al. The adsorption of aromatic macromolecules on graphene with entropy-tailored behavior and its utilization in exfoliating graphite. J. Colloid Interface Sci.2021, 599, 12-22.

R5. Deng, S.; Berry, V. Wrinkled, rippled and crumpled graphene: an overview of formation mechanism, electronic properties, and applications. Mater. Today 2016, 19, 197-212.

R6. Ling, F.; Liao, R.; Yuan, C.; Shi, X.; Li, L.; Zhou, X.; Tang, X.; Jing, C.; Wang, Y.; Jiang, S. Geometric, electronic and transport properties of bulged graphene: A theoretical study. J. Chem. Phys. 2023, 158, 8.

R7. Frisch, M.J.; Trucks, G.W.; Schlegel, H.B.; Scuseria, G.E.; Robb, M.A.; Cheeseman, J.R.; Scalmani, G.; Barone, V.; Petersson, G.A.; Nakatsuji, H.; et al. Gaussian 09, Revision A.1, Wallingford, CT, 2009.

R8. Becke, A.D. Density‐functional thermochemistry. III. The role of exact exchange. J. Chem. Phys. 1993, 98, 5648-5652.

R9. Lu, T.; Chen, F. Multiwfn: A multifunctional wavefunction analyzer. J. Comput. Chem. 2012, 33, 580-592.

Reviewer 2 Report

Comments and Suggestions for Authors

The manuscript is devoted to a computational study of the effects of cation adsorption on the structure of graphene. The authors use periodic density-functional theory calculations at the PBE level of theory to study the adsorption of Na+, K+, Mg2+, Ca2+, Cu2+ and Fe2+ on a clean graphene sheet. They study how these cations establish cation-pi (graphene) interactions and how this phenomenon in turn affects the planar structure of graphene. The aim of the work is to understand how a judicious selection of cation type can be used to induce nanometer-scale ripples on thin graphene films. The occurrence of these ripples can modify both mechanical and electronic/optical properties of graphene. The results indicate that the strength of the cation-graphene interaction determines the spatial extent of the ripples and the band gap of the adsorbed system.

The work is well carried out and focussed, and its goals are clear. The Introduction is synthetic, but provides sufficient information about why studying adsorption-induced ripple formation on graphene can have important scientific and technological implications. The computational methods used are robust and, in general, the main conclusions are well supported by the calculations. The paper could be publishable, in my opinion, but the current version suffers from a number of limitations, which I would like the authors to address.

1) How were the cations chosen? For instance, why did the authors not include also Zn2+, which has an ionic ratio very close to that of Mg2+, but a much higher electrophilic character? According to the results of the work (and to what would be expected on the basis of simple chemical considerations) the higher tendency of the Zn di-cation to attract electrons from the pi system of graphene could promote enhanced ripple formation compared to Mg2+.

2) Cu2+ and Fe3+ are magnetic ions, with unpaired electrons. Was the spin accounted for in the calculations for these systems? What were the 3d orbital occupations in the adsorbed systems?

3) The stability of the adsorbed cations should be confirmed using phonon calculations (to verify the absence of imaginary vibrational frequencies) and/or ab initio molecular dynamics simulations.

4) On page 3, the authors discuss electron transfer from graphene to the adsorbed cations. How were the electron transfer numbers estimated?

5) On page 2, the authors state that "Cu2+ demonstrates a shorter adsorption distance than both Ca2+ and Mg2+, likely due to the oxidative capabilities of copper ions." What is the origin of these enhanced oxidative capabilities? This could be understood by identifying which empty orbitals on the cation act as acceptors of electrons from graphene and what their energies are. Lower empty orbital energies enhance a cation's electron affinity. I think it would be useful to compare the energies of the lowest unoccupied orbitals of the cations, and analyze whether there is a correlation with the strength of the cation-graphene interaction.

Comments on the Quality of English Language

The paper is in general clear, although the quality of the English language could be improved. For instance, sentences like "In this study, using density functional theory simulations of attaching a metal cation [...], the tiny ripple in graphene is produced." (3rd sentence in the Abstract) should be improved. There are several similar sentences in the text that, even if understandable, may require rephrasing. 

Author Response

Referee 2:

Comments and Suggestions for Authors:

The manuscript is devoted to a computational study of the effects of cation adsorption on the structure of graphene. The authors use periodic density-functional theory calculations at the PBE level of theory to study the adsorption of Na+, K+, Mg2+, Ca2+, Cu2+ and Fe2+ on a clean graphene sheet. They study how these cations establish cation-pi (graphene) interactions and how this phenomenon in turn affects the planar structure of graphene. The aim of the work is to understand how a judicious selection of cation type can be used to induce nanometer-scale ripples on thin graphene films. The occurrence of these ripples can modify both mechanical and electronic/optical properties of graphene. The results indicate that the strength of the cation-graphene interaction determines the spatial extent of the ripples and the band gap of the adsorbed system. The work is well carried out and focused, and its goals are clear. The introduction is synthetic, but provides sufficient information about why studying adsorption-induced ripple formation on graphene can have important scientific and technological implications. The computational methods used are robust and, in general, the main conclusions are well supported by the calculations. The paper could be publishable, in my opinion, but the current version suffers from a number of limitations, which I would like the authors to address.

Reply: We sincerely thank the referee for these constructive suggestions and very positive comments on our manuscript.

(1) How were the cations chosen? For instance, why did the authors not include also Zn2+, which has an ionic ratio very close to that of Mg2+, but a much higher electrophilic character? According to the results of the work (and to what would be expected on the basis of simple chemical considerations) the higher tendency of the Zn di-cation to attract electrons from the pi system of graphene could promote enhanced ripple formation compared to Mg2+.

Reply: We thank the referee for his/her comments.

To study the effects of cation adsorption on the graphene ripple, we selected the common metal cations, such as Na+, K+, Mg2+, Ca2+, Cu2+ and Fe3+.

Now, we have added the calculation of Zn2+. As shown in Figure R3, the vertical distance of Zn2+ to graphene sheet is 9.92 Å, which is far larger than that of other cations (less than 3 Å). The ion adsorption energy (Ei) for the system with adsorption of Zn2+ is -316.2 kcal/mol, which is comparable to that of Cu2+ and about two times of that for Mg2+ and Ca2+. The numbers of transferred electrons from the graphene sheet to the unoccupied valence orbits of Zn2+ is 1.174 e, which is comparable to that of Cu2+, and about three times of that for Mg2+ and Ca2+. The value of deformation in the z direction (△Z) for rippled graphene sheet induced by the adsorption of Zn2+ is 0.004 Å, which is far less than that of Mg2+, Ca2+, Cu2+, while the relative area change of the rippled graphene compared to flat graphene (△S) is 0.61%.  Therefore, the ripple of graphene induced by Zn2+ is far smaller than that of other divalent cations such as Mg2+, Ca2+, Cu2+.

Figure R3. Structure configuration of Zn2+ adsorbed on a graphene sheet.

Changes made:

We have added the discussion and figures (Figure S4) of Zn2+ adsorption on a graphene sheet in the revised supporting information:

“As shown in Figure S4, the vertical distance of Zn2+ to graphene sheet is 9.92 Å, which is far larger than that of other cations (less than 3 Å). The ion adsorption energy (Ei) for the system with adsorption of Zn2+ is -316.2 kcal/mol, which is comparable to that of Cu2+ and about two times of that for Mg2+ and Ca2+. The numbers of transferred electrons from the graphene sheet to the unoccupied valence orbits of Zn2+ is 1.174 e, which is comparable to that of Cu2+, and about three times of that for Mg2+ and Ca2+. The value of deformation in the z direction (Z) for rippled graphene sheet induced by the adsorption of Zn2+ is 0.004 Å, which is far less than that of Mg2+, Ca2+, Cu2+, while the relative area change of the rippled graphene compared to flat graphene (S) is 0.61%.  Therefore, the ripple of graphene induced by Zn2+ is far smaller than that of other divalent cations such as Mg2+, Ca2+, Cu2+.

(2) Cu2+ and Fe3+ are magnetic ions, with unpaired electrons. Was the spin accounted for in the calculations for these systems? What were the 3d orbital occupations in the adsorbed systems?

Reply: We thank the referee for his/her comments.

We have considered the spin polarization calculations of Cu2+ and Fe3+. Their structural models have been re-optimized and the corresponding properties have been recalculated. As shown in Table S2 and Table S3, by comparing the ion adsorption energy, the number of transferred electrons, the deformation in the z direction, and the relative area change of graphene sheet before and after considering spin polarization, we find that the spin polarization has little effect on the adsorption behavior of Cu2+ onto graphene sheet, while it has a small effect on the on the adsorption behavior of Fe3+ on graphene sheet. Therefore, considering the spin polarization of Cu2+ and Fe3+, we have modified the relevant data and figures in the manuscript: For the systems with adsorption of Cu2+ and Fe3+,  the ion adsorption energies are -307.3 and -675.6 kcal/mol, the numbers of transferred electrons are1.119 and 1.677 e, the values of deformation in the z direction are 0.079 and 0.053 Å, the equivalent pressures are 5.11 and 8.70 GPa, the band gaps are 0.118 and 0.136 eV, respectively.

Table S2. The adsorption properties of Cu2+ on graphene sheet, including the ion adsorption energy (Ei), the number of transferred electrons from the graphene sheet to the unoccupied valence orbits of Cu2+ (Nt), the deformation in the z direction (△Z), the relative area change of the rippled graphene compared to flat graphene (△S), equivalent pressure (P), and bandgap (Eg).

Table S3. The adsorption properties of Fe3+ on graphene sheet, including the ion adsorption energy (Ei), the number of transferred electrons from the graphene sheet to the unoccupied valence orbits of Fe3+ (Nt), the deformation in the z direction (△Z), the relative area change of the rippled graphene compared to flat graphene (△S), equivalent pressure (P), and bandgap (Eg).

As shown in Figure R4, in the partial electron density of states (DOS) near the Fermi level of Cu2+@graphene and Fe3+@graphene, the 3d-orbital densities are much higher than those of other orbitals, and overlap with the DOS of carbon atoms at some peaks, indicating that the interactions between electrons in the 3d orbitals and on carbon atoms help the effective adsorption of transition metal cations on graphene sheets.

Figure R4. Partial electron density of states near the Fermi level of (a) Cu2+@graphene and (b) Fe3+@graphene.

Changes made:

We have modified the data and figures related to the adsorption of Cu2+ and Fe3+ in the revised manuscript, and added the discussion and new figures (Figure S5) addressing the effects of 3d orbital occupations on cation adsorption in the revised manuscript and the revised supporting information:

As shown in Figure S5, in the partial electron density of states (DOS) near the Fermi level of Cu2+@graphene and Fe3+@graphene, the 3d-orbital densities are much higher than those of other orbitals, and overlap with the DOS of carbon atoms at some peaks, indicating that the interactions between electrons in the 3d orbitals and on carbon atoms help the effective adsorption of transition metal cations on graphene sheets.

(3) The stability of the adsorbed cations should be confirmed using phonon calculations (to verify the absence of imaginary vibrational frequencies) and/or ab initio molecular dynamics simulations.

Reply: We thank the referee for his/her comments.

We have added ab initio molecular dynamics simulations using the NVT ensemble at 290 K to confirm the stability for the graphene sheets with metal adsorbed (Figure R5). After 2 ps simulation, all metal elements except Mg were observed to stay near the graphene sheets, vibrating slightly above the hollow position of the hexagonal ring on the graphene, while Mg was just a little away from the graphene sheet. This indicates that the systems of metal elements adsorbed on graphene sheets are stable.

Figure R5. Structural stability of graphene sheets with metal adsorbed after 2 ps ab initio molecular dynamics (AIMD) simulation at 290 K.

(4) On page 3, the authors discuss electron transfer from graphene to the adsorbed cations. How were the electron transfer numbers estimated?

Reply: We thank the referee for his/her comments.

The electron transfer numbers were estimated by calculating the Bader charges of all the atoms as described in the Methods section of the manuscript, and then the electron transfer numbers were obtained by comparing the electron numbers of graphene sheets before and after adsorbing the cation.

(5) On page 2, the authors state that "Cu2+ demonstrates a shorter adsorption distance than both Ca2+ and Mg2+, likely due to the oxidative capabilities of copper ions." What is the origin of these enhanced oxidative capabilities? This could be understood by identifying which empty orbitals on the cation act as acceptors of electrons from graphene and what their energies are. Lower empty orbital energies enhance a cation's electron affinity. I think it would be useful to compare the energies of the lowest unoccupied orbitals of the cations, and analyze whether there is a correlation with the strength of the cation-graphene interaction.

Reply: We thank the referee for his/her comments.

We have performed the DFT calculations with the Gaussian 09 software package [7] to analyze the Lowest Unoccupied Molecular Orbital (LUMO) energies of Mg2+, Ca2+ and Cu2+. All geometry optimizations and frequency calculations were performed at the B3LYP/SDD level of theory [8]. The LUMO equivalent surface was plotted by Multiwfn program [9].

Figure R6. The lowest unoccupied molecular orbital (LUMO) energies of Mg²⁺, Ca²⁺, and Cu²⁺ ions, indicated by cyan horizontal lines. The insets display the spatial charge distributions of the LUMOs at an isosurface value of 0.5.

The oxidative property of Cu2+ arises from its unique electronic structure, particularly the lower unoccupied orbital energy. Specifically, Cu2+ has empty 3d orbitals that are energetically accessible, facilitating electron acceptance from graphene. To quantitatively assess the relationship between the cation's oxidative capability and the interaction with graphene, we have compared the lowest unoccupied molecular orbitals (LUMOs) of Cu2+, Ca2+, and Mg2+ (Figure R6). For Mg2+, the LUMO is primarily contributed by the 3s orbital, with an energy of -18.005 eV. For Ca2+, the LUMO is mainly composed of the 4s orbital, with an energy of -14.053 eV. In contrast, the LUMO of Cu2+, predominantly involving the 3d orbital, has a much lower energy of -25.872 eV. The order of LUMO energies is Cu2+ < Mg2+ < Ca2+, which correlates well with the varying trend of adsorption energies (Cu2+ > Mg2+ > Ca2+) as described in the manuscript. This direct correlation between lower LUMO energies and stronger cation-p interactions supports the hypothesis that the enhanced oxidative property and electron affinity of Cu2+ contribute to its stronger adsorption.

Changes made:

We have added the relevant calculation method and discussion in the revised manuscript and added a new figure (Figure S6) as the same with Figure R6 in the revised supporting information:

“To analyze the Lowest Unoccupied Molecular Orbital (LUMO) energies of Mg2+, Ca2+ and Cu2+, the DFT calculations were performed with the Gaussian 09 software package [48]. All geometry optimizations and frequency calculations were performed at the B3LYP/SDD level of theory [49]. The LUMO equivalent surface was plotted by Multiwfn program [50].

The oxidative property of Cu2+ arises from its unique electronic structure, particularly the lower unoccupied orbital energy. Specifically, Cu2+ has empty 3d orbitals that are energetically accessible, facilitating electron acceptance from graphene. To quantitatively assess the relationship between the cation's oxidative capability and the interaction with graphene, we have compared the lowest unoccupied molecular orbitals (LUMOs) of Cu2+, Ca2+, and Mg2+ (Figure S6). For Mg2+, the LUMO is primarily contributed by the 3s orbital, with an energy of -18.01 eV. For Ca2+, the LUMO is mainly composed of the 4s orbital, with an energy of -14.05 eV. In contrast, the LUMO of Cu2+, predominantly involving the 3d orbital, has a much lower energy of -25.87 eV. The order of LUMO energies is Cu2+ < Mg2+ < Ca2+, which correlates well with the varying trend of adsorption energies (Cu2+ > Mg2+ > Ca2+). This direct correlation between lower LUMO energies and stronger cation-p interactions indicates that the enhanced oxidative property and electron affinity of Cu2+ contribute to its stronger adsorption.

(6) Comments on the Quality of English Language

The paper is in general clear, although the quality of the English language could be improved. For instance, sentences like "In this study, using density functional theory simulations of attaching a metal cation [...], the tiny ripple in graphene is produced." (3rd sentence in the Abstract) should be improved. There are several similar sentences in the text that, even if understandable, may require rephrasing.

Reply: We thank the referee for his/her comments.

We have carefully checked for all sentences, and made the revision in the revised manuscript:

  1. Line 3-5 in the part of Abstract, “In this study, using the density functional theory simulations of attaching a metal cation (Na+, K+, Mg2+, Ca2+, Cu2+, Fe3+) to the graphene sheet, the tiny ripple in graphene is produced.” is changed to “In this study, we report that a tiny ripple in graphene can be produced by the adsorption of a single metal cation (Na+, K+, Mg2+, Ca2+, Cu2+, Fe3+) onto graphene sheet based on the density functional theory”
  2. Line 5-6 in the part of Abstract, “We attribute this ripple to the cation-π interaction between the metal cation and the aromatic rings on the graphene surface” is changed to “We attribute this to the cation-π interaction between the metal cation and the aromatic rings on the graphene surface, which makes the carbon atoms closer to metal ions, causing the generation of the deformation of graphene sheet especially in out-of-plane direction, namely the ripple.”
  3. Line 10-12 in the part of Abstract is changed to “More importantly, the electronic and mechanical properties of graphene sheets are modified by the adsorption of various metal cations, resulting in opened bandgaps and enhanced rigidities characterized by the higher elastic modulus.”.
  4. Line 4 of the second paragraph in the part of Introduction: “wavelengths” is changed to “sizes”.
  5. Line 7 of the second paragraph in the part of Introduction: “is” is changed to “belongs to”.
  6. Line 11-12 of the second paragraph in the part of Introduction: “the type of cation should influence the ripples formed on graphene though varying the cation-p” is changed to “the type of cation should influence the ripple size formed on graphene via varying the strength of cation-p interaction.”.
  7. Line 2-3 of the second paragraph in the part of Introduction is changed to “a tiny ripple in graphene can be induced by the adsorption of metal cation.”.
  8. Line 3-8 of the third paragraph in the part of Introduction is changed to “This ripple formation is attributed to the cation-p interaction between the metal cation and the aromatic rings on the graphene surface, which makes the carbon atoms closer to metal ions, causing the generation of the deformation of graphene sheet especially in out-of-plane direction, namely the ripple. The equivalent the pressures applied to graphene sheets, which are induced by cation adsorption, are on the magnitude of GPa.”.
  9. Line 11 of the third paragraph in the part of Introduction is changed to “controlling the properties of graphene by inducing tiny ripples with metal cation adsorption.”.
  10. Line 1-2 in the part of Method: “optimization” and “was” are changed to “optimizations” and “were”, respectively.
  11. Line 11 of the first paragraph in the part of Results and discussion: “For” is changed to “As for”.
  12. Line 13-14 of the first paragraph in the part of Results and discussion is changed to “This is accessible because Mg²⁺ and Ca²⁺ carry more charges compared to Na⁺ and K⁺, respectively.”
  13. Line 15 of the first paragraph in the part of Results and discussion: “minimal” is changed to “shorter”.
  14. Line 11 of the second paragraph in the part of Results and discussion: “with the distance d” is changed to “with the varying trend of d”.
  15. Line 5 of the fourth paragraph in the part of Results and discussion: “sheet” and “number” are changed to “sheets” and “numbers”, respectively.
  16. Line 1 of the sixth paragraph in the part of Results and discussion: “effect” is deleted.
  17. Line 5 of the sixth paragraph in the part of Results and discussion: “with” is changed to “induced by”.
  18. Line 7 of the sixth paragraph in the part of Results and discussion: “graphene sheets with” is changed to “rippled graphene sheets induced by”.
  19. Line 10 of the sixth paragraph in the part of Results and discussion: “graphene” is changed to “graphene sheets”.
  20. Line 8-10 of the ninth paragraph in the part of Results and discussion is changed to “Notably, the varying trend of equivalent pressure closely follows the order of adsorption energies of various metal cations on graphene sheets.”.
  21. Line 1 of the tenth paragraph in the part of Results and discussion is changed to “The mechanical properties of rippled graphene sheets with the adsorption of various metal cations were analyzed”.
  22. The last sentence of the last paragraph in the part of Results and discussion is changed to “Therefore, the change in bandgap can be used to characterize the structural damage of graphene after deformation, as the previous study reported that the bandgaps is increased when carbon atoms are extracted from a graphene sheet in the z-direction [15].”
  23. Line 4-7 of the first paragraph in the part of Conclusion is changed to “which makes the carbon atoms closer to metal ions, causing the generation of the deformation of graphene sheet, namely the ripple. The equivalent pressures applied to graphene sheets in out-of-plane direction, generated by metal cation-π interactions, reache the magnitude of GPa.”.
  24. Line 3 of the second paragraph in the part of Conclusion, the word “significantly” is added.

References

R1. Gao, Y.; Ren, X.; Wu, J.; Hayat, T.; Alsaedi, A.; Cheng, C.; Chen, C. Graphene oxide interactions with co-existing heavy metal cations: adsorption, colloidal properties and joint toxicity. Environ. Sci.: Nano 2018, 5, 362-371.

R2. Wang, J.; Chen, B. Adsorption and coadsorption of organic pollutants and a heavy metal by graphene oxide and reduced graphene materials. Chem. Eng. J. 2015, 281, 379-388.

R3. Shi, G.; Liu, J.; Wang, C.; Song, B.; Tu, Y.; Hu, J.; Fang, H. Ion enrichment on the hydrophobic carbon-based surface in aqueous salt solutions due to cation-p interactions. Sci. Rep. 2013, 3, 3436.

R4. Meng, C.; Gao, K.; Tang, S.; Zhou, L.; Lai, W.; Luo, L.; Wang, X.; Liu, Y.; Wang, K.; Chen, Y.; et al. The adsorption of aromatic macromolecules on graphene with entropy-tailored behavior and its utilization in exfoliating graphite. J. Colloid Interface Sci.2021, 599, 12-22.

R5. Deng, S.; Berry, V. Wrinkled, rippled and crumpled graphene: an overview of formation mechanism, electronic properties, and applications. Mater. Today 2016, 19, 197-212.

R6. Ling, F.; Liao, R.; Yuan, C.; Shi, X.; Li, L.; Zhou, X.; Tang, X.; Jing, C.; Wang, Y.; Jiang, S. Geometric, electronic and transport properties of bulged graphene: A theoretical study. J. Chem. Phys. 2023, 158, 8.

R7. Frisch, M.J.; Trucks, G.W.; Schlegel, H.B.; Scuseria, G.E.; Robb, M.A.; Cheeseman, J.R.; Scalmani, G.; Barone, V.; Petersson, G.A.; Nakatsuji, H.; et al. Gaussian 09, Revision A.1, Wallingford, CT, 2009.

R8. Becke, A.D. Density‐functional thermochemistry. III. The role of exact exchange. J. Chem. Phys. 1993, 98, 5648-5652.

R9. Lu, T.; Chen, F. Multiwfn: A multifunctional wavefunction analyzer. J. Comput. Chem. 2012, 33, 580-592.

Round 2

Reviewer 1 Report

Comments and Suggestions for Authors

Dear Authors,

Thank you for considering my points. Comments and suggestions have been satisfactorily solved. I recommend moving/adapting Figures S3 and S5 to the main text.  The novelty of this work is still not clearly articulated. The introduction could benefit from refinement. Rather than relying heavily on citations, it would be more effective to better contextualize the problem and solution being addressed, providing clearer guidance for the reader. 

Comments on the Quality of English Language

Some typos, double-check text, e.g., nonzero -> non-zero

Author Response

Referee 1

Comments and Suggestions for Authors

Thank you for considering my points. Comments and suggestions have been satisfactorily solved.

We thank the referee for his/her positive comments.

  • I recommend moving/adapting Figures S3 and S5 to the main text.

Reply: We thank the referee for his/her comments.

We have moved Figure S5 to the main text as Figure 2. Figure S3 depicts the band structures and density of states (DOS) for rippled graphene sheets induced by Na+, K+, Mg2+, Ca2+, Cu2+, and Fe3+. However, we found that their band structures and DOS are quite similar. Figure 4 provides a typical band structure and density of states (DOS) for rippled graphene sheet induced by Fe3+ and encompasses the bandgaps for rippled graphene sheets with adsorption of various metal cations. Therefore, we believe that the information presented in Figure 4 is sufficient to describe the electronic properties of the rippled graphene sheets, making Figure S3 unnecessary.

Changes made:

Figure S5 has been moved to the main text as Figure 2, and we have updated the sequence numbering of all figures (highlighted in red) in the revised manuscript.

(2) The novelty of this work is still not clearly articulated. The introduction could benefit from refinement. Rather than relying heavily on citations, it would be more effective to better contextualize the problem and solution being addressed, providing clearer guidance for the reader.

Reply: We thank the referee for his/her valuable comments.

To address the novelty of this work, we have included additional discussions on the research value of tiny ripples at the nanoscale and our rationale for utilizing cation-π interactions.

Changes made:

We have added some discussions to the Introduction section and have divided the original second paragraph into two distinct paragraphs in the revised manuscript:

“Corrugation in graphene can be formed using various methods, such as spontaneous dynamic ripples of single graphene [26,27], defect-induced and doping methods [28,29], growing graphene on metal substrates [30], and graphene transfer process [31,32]. The formation of ripples at the nanoscale or smaller can create a very narrow bandgap in graphene, which is crucial for practical applications in graphene-based nanoelectronics and nanoelectromechanical devices. However, it is difficult to produce ripples smaller than one nanometer using these current methods, and accurately manipulating these ripples to achieve desired properties remains a big challenge [5,33].

The cation-π interaction, a type of non-covalent interaction formed between cations and π electron-rich carbon-based structures [34], provides a versatile and reversible approach to modifying physical properties without altering the underlying chemical structures [35]. The adsorption of cations on graphene has been observed in both gas and solvent phases [36-39]. However, no studies have investigated the effect of cation adsorption on the ripple formation in graphene. By controlling the cation adsorption site, the formation of ripples can be manipulated. Additionally, the type of cation is expected to affect the size of the ripples formed due to variations in the strength of the cation-π interactions.”

(3) Comments on the Quality of English Language.

Some typos, double-check text, e.g., nonzero -> non-zero

Reply: We thank the referee for his/her comments.

We have thoroughly checked the revised manuscript for spelling, grammatical errors, and typos, and have made the necessary revisions, which are highlighted in red:

  • Line 3 in the part of Abstract: “a tiny ripple” and “produced” are changed to “tiny ripples” and “generated”, respectively.
  • Line 7-8 in the part of Abstract is changed to “causing deformation of the graphene sheet, especially in the out-of-plane direction, thereby creating ripples.”.
  • Line 9 in the part of Abstract: “a magnitude” is changed to “magnitudes”.
  • Line 11 in the part of Abstract: “rigidities” is changed to “rigidity”.
  • Line 12 in the part of Abstract: “the” is changed to “a”.
  • Line 6 of the first paragraph in the part of Introduction: “field” is changed to “fields”.
  • Line 2 of the second paragraph in the part of Introduction: “defect or doping” is changed to “defect-induced and doping methods”.
  • Line 8-10 of the third paragraph in the part of is changed to “Additionally, the type of cation is expected to affect the size of the ripples formed due to variations in the strength of the cation-π interactions.”.
  • Line 2 of the fourth paragraph in the part of Introduction: “a tiny ripple” is changed to “tiny ripples”.
  • Line 5-6 of the fourth paragraph in the part of Introduction is changed to “causing deformation of the graphene sheet, especially in the out-of-plane direction”.
  • Line 9 of the fourth paragraph in the part of Introduction: “nonzero” is changed to “non-zero”.
  • Line 12 of the fourth paragraph in the part of Introduction: “with” is changed to “through”.
  • Line 5 of the first paragraph in the part of Methods: “interaction” and “function” are changed to “interactions” and “functions”, respectively.
  • Line 7 of the first paragraph in the part of Methods is changed to “the Perdew–Burke–Ernzerhof (PBE) functional”.
  • Line 12 of the first paragraph in the part of Methods is changed to “For Cu2+ and Fe3+, which have unpaired electrons”.
  • Line 1 of the second paragraph in the part of Methods is changed to “lowest unoccupied molecular orbital (LUMO)”.
  • Line 2 of the first paragraph in the part of Results and discussion is changed to “we initially constructed structural models”.
  • Line 6 of the first paragraph in the part of Results and discussion: “are positioned at” is changed to “occupy”.
  • Line 10 of the first paragraph in the part of Results and discussion: “observation” is added.
  • Line 11-12 of the first paragraph in the part of Results and discussion is changed to “Na+ and Mg2+ compared to K+ and Ca2+, respectively. Additionally, when comparing metal ions”.
  • Line 13-14 of the first paragraph in the part of Results and discussion is changed to “This trend can be attributed to the higher charges of Mg²⁺ and Ca²⁺ relative to Na⁺ and K⁺, respectively.”.
  • Line 3 of the fourth paragraph in the part of Results and discussion: “find” is changed to “found”.
  • Line 4 of the fourth paragraph in the part of Results and discussion: “numbers” is changed to “number”.
  • Line 2 of the fifth paragraph in the part of Results and discussion: “much” is changed to “considerably”.
  • Line 4 of the sixth paragraph in the part of Results and discussion is changed to “These parameters help describe”.
  • Line 5 of the sixth paragraph in the part of Results and discussion: “for” is changed to “corresponding to”.
  • Line 2-3 of the seventh paragraph in the part of Results and discussion is changed to “the previously flat graphene structure becomes destabilized, as it can no longer maintain its lowest free energy state.”.
  • Line 5-6 of the seventh paragraph in the part of Results and discussion is changed to “Thus, the deformation particularly in the out-of-plane direction, namely the ripple, occurs as a direct result of the adsorption.”.
  • Line 1-2 of the eighth paragraph in the part of Results and discussion is changed to “we introduce the concept of equivalent pressure P”.
  • Line 19 of the eighth paragraph in the part of Results and discussion is changed to “When the corrugation is isotropic (i.e., forming ripples)”.
  • Line 21 of the eighth paragraph in the part of Results and discussion is changed to “used in this study to calculate equivalent pressure”.
  • Line 6-7 of the ninth paragraph in the part of Results and discussion is changed to “the elastic modulus of the rippled graphene increases with the presence of these cations.”.
  • Line 3 of the tenth paragraph in the part of Results and discussion is changed to “a zero bandgap (0 eV)”.
  • Line 6-7 of the tenth paragraph in the part of Results and discussion is changed to “(Figure 4b and Figure S3). The adsorption of these metal cations leads to the opening of energy bandgaps in graphene sheets”.
  • Line 8 of the tenth paragraph in the part of Results and discussion: “nonzero” is changed to “non-zero”.
  • Line 10 of the tenth paragraph in the part of Results and discussion: “Therefore” is changed to “Consequently”.
  • Line 11-12 of the tenth paragraph in the part of Results and discussion is changed to “as previous study reported an increase in bandgap”.
  • Line 5-6 of the first paragraph in the part of Conclusion is changed to “inducing deformation of the graphene sheet, referred to as the ripple”.
  • Line 2 of the second paragraph in the part of Conclusion is changed to “we anticipate that”.
  • Line 3-5 of the second paragraph in the part of Conclusion is changed to “Therefore, we aim to explore the possibility of obtaining a graphene sheet with a wide bandgap by adsorbing multiple metal cations—this will be a primary focus for our future research.”.
  • Line 7 of the second paragraph in the part of Conclusion: “based on” is changed to “leveraging”.
  • Reference 51 in original version is deleted.

Reviewer 2 Report

Comments and Suggestions for Authors

The authors have addressed very carefully and convincingly all the points raised in my previous report. The paper has been improved substantially and it is now in my opinion suitable for publication in Nanomaterials.

Comments on the Quality of English Language

In the revised paper, the authors have addressed several text inaccuracies appearing in the original submission. Minor text changes and spell checking may be required before publication.

Author Response

Referee 2

Comments and Suggestions for Authors

The authors have addressed very carefully and convincingly all the points raised in my previous report. The paper has been improved substantially and it is now in my opinion suitable for publication in Nanomaterials.

Comments on the Quality of English Language

In the revised paper, the authors have addressed several text inaccuracies appearing in the original submission.

We thank the referee for his/her positive comments.

  • Minor text changes and spell checking may be required before publication.

Reply: We thank the referee for his/her comments.

We have thoroughly checked the revised manuscript for spelling, grammatical errors, and typos, and have made the necessary revisions, which are highlighted in red:

  • Line 3 in the part of Abstract: “a tiny ripple” and “produced” are changed to “tiny ripples” and “generated”, respectively.
  • Line 7-8 in the part of Abstract is changed to “causing deformation of the graphene sheet, especially in the out-of-plane direction, thereby creating ripples.”.
  • Line 9 in the part of Abstract: “a magnitude” is changed to “magnitudes”.
  • Line 11 in the part of Abstract: “rigidities” is changed to “rigidity”.
  • Line 12 in the part of Abstract: “the” is changed to “a”.
  • Line 6 of the first paragraph in the part of Introduction: “field” is changed to “fields”.
  • Line 2 of the second paragraph in the part of Introduction: “defect or doping” is changed to “defect-induced and doping methods”.
  • Line 8-10 of the third paragraph in the part of is changed to “Additionally, the type of cation is expected to affect the size of the ripples formed due to variations in the strength of the cation-π interactions.”.
  • Line 2 of the fourth paragraph in the part of Introduction: “a tiny ripple” is changed to “tiny ripples”.
  • Line 5-6 of the fourth paragraph in the part of Introduction is changed to “causing deformation of the graphene sheet, especially in the out-of-plane direction”.
  • Line 9 of the fourth paragraph in the part of Introduction: “nonzero” is changed to “non-zero”.
  • Line 12 of the fourth paragraph in the part of Introduction: “with” is changed to “through”.
  • Line 5 of the first paragraph in the part of Methods: “interaction” and “function” are changed to “interactions” and “functions”, respectively.
  • Line 7 of the first paragraph in the part of Methods is changed to “the Perdew–Burke–Ernzerhof (PBE) functional”.
  • Line 12 of the first paragraph in the part of Methods is changed to “For Cu2+ and Fe3+, which have unpaired electrons”.
  • Line 1 of the second paragraph in the part of Methods is changed to “lowest unoccupied molecular orbital (LUMO)”.
  • Line 2 of the first paragraph in the part of Results and discussion is changed to “we initially constructed structural models”.
  • Line 6 of the first paragraph in the part of Results and discussion: “are positioned at” is changed to “occupy”.
  • Line 10 of the first paragraph in the part of Results and discussion: “observation” is added.
  • Line 11-12 of the first paragraph in the part of Results and discussion is changed to “Na+ and Mg2+ compared to K+ and Ca2+, respectively. Additionally, when comparing metal ions”.
  • Line 13-14 of the first paragraph in the part of Results and discussion is changed to “This trend can be attributed to the higher charges of Mg²⁺ and Ca²⁺ relative to Na⁺ and K⁺, respectively.”.
  • Line 3 of the fourth paragraph in the part of Results and discussion: “find” is changed to “found”.
  • Line 4 of the fourth paragraph in the part of Results and discussion: “numbers” is changed to “number”.
  • Line 2 of the fifth paragraph in the part of Results and discussion: “much” is changed to “considerably”.
  • Line 4 of the sixth paragraph in the part of Results and discussion is changed to “These parameters help describe”.
  • Line 5 of the sixth paragraph in the part of Results and discussion: “for” is changed to “corresponding to”.
  • Line 2-3 of the seventh paragraph in the part of Results and discussion is changed to “the previously flat graphene structure becomes destabilized, as it can no longer maintain its lowest free energy state.”.
  • Line 5-6 of the seventh paragraph in the part of Results and discussion is changed to “Thus, the deformation particularly in the out-of-plane direction, namely the ripple, occurs as a direct result of the adsorption.”.
  • Line 1-2 of the eighth paragraph in the part of Results and discussion is changed to “we introduce the concept of equivalent pressure P”.
  • Line 19 of the eighth paragraph in the part of Results and discussion is changed to “When the corrugation is isotropic (i.e., forming ripples)”.
  • Line 21 of the eighth paragraph in the part of Results and discussion is changed to “used in this study to calculate equivalent pressure”.
  • Line 6-7 of the ninth paragraph in the part of Results and discussion is changed to “the elastic modulus of the rippled graphene increases with the presence of these cations.”.
  • Line 3 of the tenth paragraph in the part of Results and discussion is changed to “a zero bandgap (0 eV)”.
  • Line 6-7 of the tenth paragraph in the part of Results and discussion is changed to “(Figure 4b and Figure S3). The adsorption of these metal cations leads to the opening of energy bandgaps in graphene sheets”.
  • Line 8 of the tenth paragraph in the part of Results and discussion: “nonzero” is changed to “non-zero”.
  • Line 10 of the tenth paragraph in the part of Results and discussion: “Therefore” is changed to “Consequently”.
  • Line 11-12 of the tenth paragraph in the part of Results and discussion is changed to “as previous study reported an increase in bandgap”.
  • Line 5-6 of the first paragraph in the part of Conclusion is changed to “inducing deformation of the graphene sheet, referred to as the ripple”.
  • Line 2 of the second paragraph in the part of Conclusion is changed to “we anticipate that”.
  • Line 3-5 of the second paragraph in the part of Conclusion is changed to “Therefore, we aim to explore the possibility of obtaining a graphene sheet with a wide bandgap by adsorbing multiple metal cations—this will be a primary focus for our future research.”.
  • Line 7 of the second paragraph in the part of Conclusion: “based on” is changed to “leveraging”.
  • Reference 51 in original version is deleted.
